# Changes in the Microbial Composition of the Cecum and Histomorphometric Analysis of Its Epithelium in Broilers Fed with Feed Mixture Containing Fermented Rapeseed Meal

**DOI:** 10.3390/microorganisms9020360

**Published:** 2021-02-12

**Authors:** Ida Szmigiel, Damian Konkol, Mariusz Korczyński, Marcin Łukaszewicz, Anna Krasowska

**Affiliations:** 1Department of Biotransformation, Faculty of Biotechnology, University of Wroclaw, ul. F. Joliot-Curie 14a, 50-383 Wrocław, Poland; ida.szmigiel@gmail.com (I.S.); marcin.lukaszewicz@uwr.edu.pl (M.Ł.); 2Department of Environment Hygiene and Animal Welfare, Faculty of Biology and Animal Sciences, Wrocław University of Environmental and Life Science, Chełmońskiego 38C, 51-630 Wrocław, Poland; damian.konkol@upwr.edu.pl (D.K.); mariusz.korczynski@upwr.edu.pl (M.K.)

**Keywords:** rapeseed meal, probiotics, *Bacillus subtilis*, broilers

## Abstract

This study examined the influence of fermented rapeseed meal (FRSM) on the intestinal morphology and gut microflora of broiler chickens. Limited information is available on the effects of FRSM on the intestinal morphology and the gastrointestinal microbiome population of animals. First, 48 21-day Ross 308 broilers were placed in metabolic cages and randomly assigned to four experimental groups. Group I birds were negative controls and received no additive. Group II birds were positive controls and received a 3% addition of unfermented rapeseed meal. Group III birds received a 3% addition of rapeseed meal fermented with the *Bacillus subtilis* 67 bacterial strain. Group IV birds received a 3% addition of rapeseed meal fermented with the *B. subtilis* 87Y strain. After 23 days of experimental feeding, the contents of the birds’ ceca were collected for microorganism determination. The histomorphology of the broilers’ ceca was also determined, and beneficial changes were found in the histology of the broilers’ ceca with the additives. Moreover, these materials inhibited the growth of pathogens and significantly stimulated the growth of probiotic bacteria. These results suggest that the addition of 3% FRSM has a potential probiotic effect and can be used as a material in feed for broilers.

## 1. Introduction

The gastrointestinal microbiome of chickens is very diverse and consists of over 900 species of microorganisms [1]. It affects the production parameters of broilers and plays an essential role in protecting against pathogens, detoxifying and modulating the immune system [2,3]. The composition of the gut microbiome varies depending on the section of the gastrointestinal tract. Evidence suggests that the cecum is important to chickens’ health and is a major reservoir for pathogens [3]. The cecum is also an essential site for the fermentation, transportation, and absorption of nutrients, potentially influencing chicken productivity [4,5]. As a result, the composition of the cecal microbiome is increasingly being studied. Thus, it is necessary to develop methods that alter the poultry gastrointestinal tract microbiome to improve the economic results of production.

The byproducts of food processing, such as rapeseed meal (RSM), can be used for fermentation, resulting in fermented RSM (FRMS) that contains beneficial nutrients for poultry feeding, such as a high amount of sulfur amino acids, but the absorption of these compounds is limited due to their high fiber content [6,7]. To date, studies have proved that fermentation with microorganisms such as *Bacillus subtilis, Candida utilis,* or *Enterococcus faecalis* can lead to the significant degradation of crude fiber and other antinutritional compounds [8].

The probiotic bacteria used during the fermentation of rapeseed meal may have positive effects on performance, animal health, quality of animal products, digestive enzyme activity, and intestinal morphology [9,10,11,12]. Moreover, solid-state fermentation (SSF) with an RSM base leads to high value-added products such as fodder enzymes, polymers, or biosurfactants [13,14].

In this study, the *Bacillus subtilis* 87Y strain isolated from the earthworm *Eisenia fetida* was used for the fermentation of RSM [15]. We previously showed that the *B. subtilis* 87Y strain has a wide range of enzymatic activities and is a great producer of the lipopeptide surfactin. *B. subtilis* 87Y has shown probiotic properties during in vitro studies by promoting *Lactobacillus* strains and simultaneously inhibiting *Salmonella* strains [15]. However, there is limited research on the effects of FRSM on cecum morphology and on the ceca microbiome population. Therefore, in this study, we investigated the effect of FRSM using the *B. subtilis* 87Y and 67 strains on the microbial composition in the cecum and the morphology of the ceca of broiler chickens.

## 2. Materials and Methods

### 2.1. Fermentation of Rapeseed Meal by the Bacillus subtilis 87Y and 67 Strains in 50 kg SSF Bioreactor

The *B. subtilis* 87Y and *B. subtilis* 67 strains, both isolated from the earthworm *Eisenia fetida* [15], were used for the fermentation of RSM. Bacterial inoculum with an optical density OD_600_ = 0.1 (0.35 × 10^7^ cfu/mL) was prepared by resuspending the *B. subtilis* precultures in minimal MIM1 medium (sucrose 60 g/L (LabEmpire, Rzeszów, Poland), urea 2.3 g/L (LabEmpire, Rzeszów, Poland), MgSO_4_ 0.5 g/L (LabEmpire, Rzeszów, Poland), Na_2_HPO_4_ 8.4 g/L (Pol-Aura, Dywity, Poland), NaH_2_PO_4_ 3.9 g/L (Pol-Aura, Dywity, Poland), FeSO_4_ 1.2 mg/L (Pol-Aura, Dywity, Poland), CuSO_4_ 1.6 mg/L (Pol-Aura, Dywity, Poland), MnSO_4_ 5 mg/L (Pol-Aura, Dywity, Poland), and pH = 7.0). RSM was pasteurized for 8 min at 80 °C and then mixed with prepared inoculum in a 1:1 ratio (m/v) to maintain 50% humidity. Fermentation was performed with 20 rpm shaking and 50 L/min aeration for 24 h at 37 °C. After 24 h, the RSM was immediately dried using a fluid bed drier. The *B. subtilis* concentration (in the case of both strains) in the fermented additive was 1.8 × 10^7^ cfu/g.

### 2.2. Broiler Chicken Population and Experimental Design

This research was carried out in an experimental henhouse at the Agricultural Experimental Station “Swojec” in Wroclaw, Poland. A total of 48 Ross 308 broiler chickens (21 days of life, 978.58 g, SD ± 4.2 of live weight) were used for this study. The birds were kept in controlled microclimatic conditions according to the breeder’s recommendations. The setpoint for temperature was 32 °C for day 1 and slowly reduced to 22 °C through day 21, and then held constant until day 44. In the first week, there were 10 min of darkness every 6 h; after the seventh day, there were 18 h of light and 6 h of darkness, and in the last three days of fattening, light was provided for 24 h [16]. In their 21 days of life, the chickens were placed in metabolic cages. The birds were divided into four experimental groups with each group consisting of 12 birds. Each group had four replicates, and each replicate consisted of three birds. Group I birds were negative controls (NCs) and received no additive. Group II birds were positive control (PCs) and received 3% addition of unfermented rapeseed meal (URSM). Group III (67) birds received a 3% addition of FRSM with the bacterium *B. subtilis* 67. Group IV (87Y) birds received a 3% addition of rapeseed meal fermented with *B. subtilis* 87Y. The addition of 3% RSM was due to the nutritional standards for poultry by Smulikowska [17]. Rapeseed meal was used in this study only as a carrier for the meal-fermenting bacteria—it was not used as a substitution of other feed components. The fermented rapeseed meal we obtained was registered with the veterinary identification number PL0461097p.

### 2.3. Nutrition

The birds were fed in accordance with the Poultry Nutritional Standards [17]. Table 1 presents the composition of the basal diet. The feed was provided in the form of pellets. Birds received food and water ad libitum. The feed intake was monitored in all groups and was in accordance with the line manufacturer’s instructions [16], as presented in Appendix A.

### 2.4. Sampling to Determine the Microbiome of Broiler Chickens

Four chickens from each group were euthanized at the beginning of the experiment and after 44 days of life. The hens were euthanized by a percussive blow to the head and then exsanguinated in accordance with the Directive 2010/63/EU of the European Parliament and of the Council of 22 September 2010 on the protection of animals used for scientific purposes. After exsanguination, the ceca of each bird was removed, and the contents were collected. The broiler ceca contents from each research group (NCs, PCs, 67, and 87Y) were aseptically sampled, mixed with 50% glycerol, and frozen at–80 °C immediately after preparation.

### 2.5. Detection and/or Quantification of Cecal Microorganisms

Chromogenic media were used for the initial analysis of the broiler microbiome. Detection of pathogens such *Salmonella* spp. or *Klebsiella* spp. was performed using RAPID *Salmonella* medium (Bio-Rad, California, USA). Probiotic bacteria such as *Lactobacillus* spp. were detected and quantified using StrepB Select medium (Bio-Rad, CA, USA). The plating of cecal material on chromogenic agar was carried out using ISO standards. Pathogen detection was performed according to PN-EN ISO 6579-1 (horizontal method for the detection, enumeration, and serotyping of *Salmonella*—Part 1: Detection of *Salmonella* spp.) with modifications. Briefly, buffered peptone water (BPW) was inoculated with the cecal sample (tenfold dilution) and incubated for 18 h at 37 °C. Next, 100 µL of inoculated BPW was transferred to 10 mL of RVS broth and incubated for 24 h at 41.5 °C. Finally, the amplified cecal sample was plated on RAPID *Salmonella* agar in triplicate. Moreover, raw, unamplified material was plated on RAPID *Salmonella* agar in serial dilutions for the quantification of colonies. The detection of *Lactobacillus* spp. was performed according to PN ISO 15214:2002 (horizontal method for the enumeration of mesophilic lactic acid bacteria—colony-count technique at 30 °C) with some modifications. Briefly, sample dilutions with BPW were performed according to ISO 6887-1:2017 and were then plated on StrepB select agar and MRS (Sigma-Aldrich, St. Louis, CA, USA) agar in serial dilutions and quantified. Detected colored colonies on both chromogenic media were then collected for identification using MALDI-TOF-MS.

### 2.6. MALDI-TOF-MS Analysis

The identification of cecum microbiota was determined using MALDI-TOF-MS. For analysis, five colonies were taken from each morphology found within the plated material. Preparation of the microorganisms was performed according to the manufacturer’s instructions. Briefly, single colonies were mixed with 300 µL distilled water and thoroughly resuspended. Next, 900 µL of 100% ethanol was added and mixed well. Probes were then centrifuged at 13,000 rpm for 2 min, and the supernatant was removed. Next, 50 µL of 70% formic acid was added to the pellet, mixed thoroughly, and a proportional amount of acetonitrile was added. After centrifugation at 20,000× *g* for 2 min, 1 µL of supernatant was spotted onto a MALDI plate (384 MTP Polished Steel, Bruker Daltonics, Billerica, MA, USA), dried, overlayed with 1 µL of α-cyano-4-hydroxycinnamic acid (HCCA) matrix solution, and examined by MALDI-TOF-MS. Mass spectra were generated by a ultrafleXtreme MALDI-TOF mass spectrometer (Bruker Daltonics, Billerica, MA, USA) equipped with a nitrogen laser (λ = 337 nm) operating in linear positive ion detection mode under flexControl 3.4 (Bruker Daltonics, Billerica, MA, USA). The results are shown, along with confidence scores ranging from 0.00 to 3.00.

### 2.7. Metagenome Analysis

The total ceca contents of the birds were processed in pools. One pool contained four cecal digestion samples from each group: 21_NC, 44_NC, 44_PC, 44_67, and 44_87. Samples from each group were frozen in liquid nitrogen and kept at −80 °C. Metagenomic analysis was performed by Eurofins Genomics (Ebersberg, Germany) and included cell lysis, DNA purification, and quantification. A random DNA library was prepared from the whole microbiome. Sequencing of the whole library was performed with Illumina HiSeq2500 technology and a paired end-run type. The read length was 2 × 150 bp. The results are provided as operational taxonomic units (OTUs) abundance, which are classified by Kraken’s exact k-mer alignment algorithm and the appropriate MiniKrakenDB database. Taxonomic profiling was done using KrakenUniq and the Minikraken reference database [18] and using the National Center for Biotechnology Information (NCBI) databases of bacterial, archaeal, fungal, protozoan, and viral genomes.

### 2.8. Histomorphometrics of the Cecum

The cecal samples (from both cecal deviations and proximal regions) were fixed in 4% buffered formalin and embedded in paraffin blocks. Then, 5 µm paraffin sections obtained on a Hyrax M25 (Carl Zeiss, Oberkochen, Germany) rotational microtome were stained with hematoxylin and eosin (HE) according to the histological protocol. An intestinal cross-sectional analysis was performed using an optical microscope (Axio Scope^®^ A1; Carl Zeiss, Oberkochen, Germany). Photographs of the villi were collected, and their height and width, the distance between the villi, and the number of villi were measured using Image J (LOCI, University of Wisconsin, Madison, WI, USA). The mean was calculated based on 15 measurements from each sample in three replicates; a quarter of the field of view of the image was used when making measurements. In the description of the morphological images, the I See Inside (ISI) method was used [19].

### 2.9. Statistical Analyses

Data related to the content of the microbiota were logged and presented as log10 cfu/mg of feces. Data normality was assessed using the Shapiro–Wilk test. If the distribution was normal, then a one-way analysis of variance was performed followed, by Tukey’s post-hoc test. If the distribution was not normal, then the Kruskal–Wallis test was performed. Differences were considered statistically significant when *p* < 0.05. Data were analyzed using Statistica ver. 13.1.

## 3. Results

### 3.1. Detection of Microorganisms Using RAPID Salmonella agar and Identification by MALDI-TOF-MS

The PN-EN ISO 6579-1 standard (horizontal method for the detection, enumeration, and serotyping of *Salmonella*—Part 1: Detection of *Salmonella* spp.) was used, along with RAPID *Salmonella* agar chromogenic media. Within the plated cecal material, we observed violet, green, white, and dark blue colonies (Appendix A in Appendix A). The reference bacterial strains plated on RAPID *Salmonella* did not give undeniable results in the color-dependent species identification. Hence, we decided to confirm the identification of microorganisms using MALDI-TOF-MS (Table 2). The MALDI analysis revealed that the violet colonies were *Salmonella* spp., the green colonies were *Enterobacter cloacae*, the blue colonies were *Klebsiella pneumoniae*, and the white colonies were *Escherichia coli* (Table 2).

We did not observe any *Salmonella* colonies within the plated the raw and unamplified cecal material. However, amplification of the cecal content in accordance with the ISO standard did show *Salmonella* spp. in one of the four chickens from samples taken at the start of the experiment (day: 21; group: NC; Table 3) but not in any group of chickens from day 44 (Table 3).

Moreover, we observed that 50% of the negative control group (day: 44; group: NC) developed *K. pneumoniae* after 44 days (Table 3). Chickens fed with 3% of URSM (PC), as well as with 3% of FRSM (67 and 87Y), were not infected by *K. pneumoniae* (Table 3). Next, green *Enterobacter* colonies showed up within the samples. We observed *E. cloacae* within three of the four cecal samples from chickens fed without additive (day: 44; group: NC), as well as within three of the four cecal samples from chickens fed with 3% addition of URSM (day: 44; group: PC)—both were from the last day of the experiment. Groups fed with FRSM were not infected by *E. cloacae* (Table 3).

Finally, white colonies, later identified as *E. coli*, were present in almost every amplified broiler cecum. They were also the only ones we observed and quantified in raw seeded material on RAPID *Salmonella* agar (Table 4). The number of *E. coli* found in the raw material from the first to the last day of the experiment doubled in the negative control group (NC). On day 44, the abundance of *E. coli* was slightly different in the groups fed with additives (PC, 67, and 87Y) (Table 4).

### 3.2. Microorganism Detection by Chromogenic Agar Media and Identification by MALDI-TOF-MS: Detection of Microorganisms Using StrepB Select Agar

Several morphologically and differently colored colonies were observed in the material spread on StrepB medium (Appendix A). The colonies were classified as blue, lilac, violet matte, violet/slightly pink, and light violet (Appendix A). All of the observed colonies were counted. Each type of colony was observed in each trial.

PN ISO 15214_2002P also provides guidance on the use of a reference medium in terms of which material should be shown in parallel to confirm the results. MRS medium (DeMan, Rogosa, and Sharpe medium) is dedicated to the cultivation of *Lactobacillus* bacteria. Three types of colonies were observed in the material spread on MRS agar: Large white, matte white, and transparent (Appendix A). Species identification of colonies grown on StrepB select agar and MRS agar (Table 5) was also performed using MALDI-TOF-MS.

Among the *Lactobacilli* species, we found *L. gasseri, L. salivarus, L. reuterii, L. crispatus*, and/or *L. johnsonii* (Table 5). After 44 days, the amount of *Lactobacillus* spp. dramatically decreased in the negative control group (day: 44; group: NC) (Table 4). The other experimental groups maintained the quantity of probiotics (Table 4), except for two incidents in chickens fed with FRSM (day: 44; group: 67 and 87Y) when the amount of *L. salivarus* and *L. reuterii* decreased to 0, respectively. Interestingly, in group 67, *L. salivarus* was detected only on MRS agar and not on StrepB agar, although *L. salivarus* was detected in both media. StrepB agar also showed the presence of *E. faecalis* and *E. faecium*. The quantity of *E. faecalis* in the chicken ceca remained mostly stable.

### 3.3. Metagenome Analysis

Most of the microorganisms found in the broiler cecal material could not be identified or classified. The identified taxonomic data were a maximum of only approximately 30% of the material content (Table 6).

Broilers fed with 3% addition FRSM had an increased bacteria population in their cecal material. Feed without any additional material had a significantly increased abundance of *Eukaryote*, starting from 0.13% in the control group (21_NC) to 6% at the end of the experiment (44_NC). Simultaneously, the addition of URSM or FRSM (44_67 or 44_87Y) decreased the abundance of *Eukaryote* to 0.05%, 0.04%, and 0.03%, respectively (Table 7). The *Archaea* kingdom increased 10-fold in the negative control group after feeding and remained unaffected in the rest of the research groups. The fungi and viruses did not change during the experiment (Table 7).

The *Bacteroidetes* phylum occupied the majority (approximately 80%) of the identified cecal material among all of the samples. URSM or FRSM addition led to *Bacteroidetes* at the same level; however, we did observe a 50% decrease of this phylum in the negative control (44_NC) (Figure 1). The second-most abundant phylum was *Firmicutes*, sharing approximately 25% of the found microorganisms, with a slight decrease only in the cecal samples from birds fed with *B. subtilis* 87Y FRSM. Many *Proteobacteria* were found at the end of the experiment in the negative control group (Figure 1).

The *Bacteroidetes* phylum represented the *Alistipes* and *Bacteroides* genera (Figure 2). *Alistipes* slightly increased after 23 days of feeding with the *B. subtilis* 87Y strain (23_87) FRSM. A drop to 60% of the content of *Alistipes* was noticeable after feeding with URSM and fermented with the *B. subtilis* 67 strain, and up to 40% with additional materials being free fed. Significant changes were observed in the genus *Bacteroides.* From a small initial amount of these bacteria, their content increased to 20% (44_NC) after feeding without additional materials. The addition of strains 67 or 87Y (44_67, 44_87Y) to FRSM resulted in an approximately 25% content of *Bacteroides*. Only a slight increase of this genus was observed in the URSM sample (44_PC) (Figure 2).

The predominant species in the *Alistipes* genus was *Alistipes finegoldii,* and it initially occupied approximately 80% of the population of microorganisms belonging to the *Alistipes* genus (Figure 3). The addition of strain 87Y to FRSM slightly increased the proportion of *A. finegoldii*. The addition of strain 67 to URSM and FRSM resulted in a decrease in the content of this species to approximately 60%, while feeding with no additional materials led to a ~45% decrease in the chickens’ cecal content. A decrease in *A. finegoldii* in samples 44_NC, 44_PC, and 44_67 was associated with a significant increase in *Bacteroides vulgatus*—occupying only approximately 5% in sample 44_87Y, approximately 15% in 44_NC, and approximately 20% in 44_PC and 44_67. We also observed the *Blastocystis hominis* (approximately 6%) parasite after feeding with no additional materials (44_NC). This *Eukaryote* did not develop upon the addition of URSM or FRSM to the feed (samples 44_PC, 44_67, and 44_87Y) (Figure 3).

### 3.4. Histomorphometric Analysis

The height, width, and distance between the villi and the number of villi were measured to determine the morphometry of the ceca. The addition of 3% URSM or FRSM caused a slight decrease in the height and width of the cecal villi, along with the amount of villi (Appendix A).

Significant deformations of the cecal mucosa were observed in the negative control birds (day: 21; group: NC) receiving no URSM or FRSM. The bases were enlarged, and the villi were elongated with thin tops and thickened brush seams. The presence of lymphocytes in the base and on the edges of the villi may indicate increased immunological activity. The crypts were elongated and occasionally arranged in two layers. In dilated villi bases, the crypts were open to the intestinal lumen, which most likely lacked crypts or had damaged epithelium in these places. We also observed apoptotic, stratified, and blunt upper parts of the villi (Figure 4).

The addition of a 3% URSM (day: 44; group: PC) did cause noticeable changes to the intestinal mucosa (Figure 5). The brush limb was visible, along with the entire height of the villi, with the undisturbed vascular system visible in the middle. Only a few lymphocytes were present on the edges of the villi. Crypts were ovular and closed to the intestinal lumen. The villi formed a half-wave structure that was slightly inclined toward the base of the intestinal musculature. Although the inter-villus spaces were slightly reduced, the inclination of the villi was natural and did not reduce tissue functionality (Figure 5).

A wave-like villi structure was observed in both groups receiving the 3% addition of FRSM (day: 44; group: 67 and 87Y). Densely arranged crypts in several layers were also observed (Figure 6 and Figure 7). Single apoptotic villi were observed in the 67 group (Figure 6), and visible goblet cells were seen at the edges of the villi in the 87Y group (Figure 7). A brush seam was present over the entire length of the villi (day: 44; group: 67) or over three quarters of the length of the villi (day: 44; group: 87Y). The reduction of goblet cells resulted in increased adhesion of pathogens to the mucosa [20]. Goblet cells were visible in the mucosa of birds fed with 3% FRSM additive in group 87Y.

## 4. Discussion

The aim of this study was to investigate the possible positive effects of fermented rapeseed meal as a feed supplement on the broiler microbiome.

We used two approaches to determine the changes taking place in the cecal microbiome of chickens fed with fermented rapeseed meal. First, according to ISO standards and chromogenic media, we initially determined the selected probiotics and pathogens in broiler ceca, as confirmed using MALDI-TOF/MS. We then used metagenomics data to estimate and determine the whole microbiome, including non-culturable species.

Among the material plated on RAPID *Salmonella* media, we found some well-known pathogens of poultry. *Salmonella* is one of the primary cause of foodborne illness and remains a worldwide public health concern [21,22,23]. An infectious dose in healthy humans is approximately 10^6^–10^8^ cfu/mL, but lower bacterial counts may cause diseases under certain conditions [24].

Another pathogen identified was *Klebsiella*, an opportunistic pathogen that can cause infectious diseases, such as urinary tract infections and pneumonia, and is also associated with increased morbidity and mortality [25,26]. Hamza et al. showed that commercial broiler slaughter plants might contain multidrug-resistant *Klebsiella* strains [27]. Another increasingly dangerous human pathogen found within the material was *Enterobacter*, which is responsible for 4%–12% of cases of bacteremia, especially for respiratory and urinary tract infections [28]. Studies have shown that the presence of various species of *Enterobacteriaceae* on eggshells is problematic [29,30,31]. However, as previously described, pathogens were only observed on RAPID *Salmonella* agar within amplified material; thus, we cannot declare the exact amount of these microorganisms. Importantly, *Salmonella* was found just inside one of four broiler chickens at the start of the experiment. The other pathogens mentioned herein appeared after 23 days of feeding, although only in the negative or positive control groups (fed without additive or with 3% of URSM, respectively).

The decreases in *Salmonella* spp. maybe related to an acidifying foregut during fermented feed (FF) consumption [32]. *E. coli* was also found in the broiler ceca. Many *E. coli* strains are harmless commensals naturally found in the intestinal tract, but some can cause intestinal or extraintestinal diseases, such as extraintestinal pathogenic *E. coli* (ExPEC) [33]. *E. coli* was the only species that showed up within raw plated material on RAPID *Salmonella* agar. The quantity of the *E. coli* colonies did not change significantly during the experiment. StrepB agar showed two more pathogen species (*E. faecalis* and *E. faecium*) that were not observed on MRS or RAPID *Salmonella* media.

*Enterococci* are frequent contaminants of poultry meat and well-established pathogens in human medicine. They may result in issues ranging from uncomplicated wound infections to fatal endocarditis [34,35]. Colonies of interesting *S. alactolyticus* also appeared on StrepB agar. *S. alactolyticus* is a lactic acid bacteria and has been found to be the dominant species isolated from canine feces [36]. However, infections of *S. alactolyticus* can cause bacteremia by infective endocarditis. Moreover, antibiotic-resistant strains have been found in raw milk [37,38,39]. The quantity of these three opportunistic pathogens (*E. faecalis, E. faecium*, and *S. alactolyticus*) changed slightly, although insignificantly, during the time of the experiment.

Taking into account the results so far, we came to the conclusion that the addition of 3% FRSM fermented with *B. subtilis* strains inhibits the development of pathogens. Similar results were obtained by Manafi, who also observed a *Salmonella* reduction after feeding broilers with a *Bacillus* species mixture [40]. This result is also compatible with our previous study, where we showed a significant decrease of *Salmonella* in in vitro co-culture with *B. subtilis* 87Y. In turn, Buahom suggested that survivor probiotic bacteria are due to the induction of competitive exclusion (CE) by inhibition of the attachment of pathogenic microbes to the gastrointestinal tract [41].

Lactic acid bacteria (LAB) are mainly found on MRS agar. They maintain a microbial balance and protect against pathogenic infections [10,42]. They also improve digestion and nutrient assimilation, remove toxic substances, and enhance immunity [43,44]. Most of the experimental groups maintained the quantity of probiotics, except for two incidents in chickens fed with FRSM (day: 44; group: 67, 87Y). This result sums up with a noticeable decrease in LAB at the end of the experiment in the negative control. We can also consider this as a bacteriostatic effect. However, *Bacillus* species have been shown to increase the *Lactobacilli* population within the gastrointestinal tract of poultry [45].

Chromogenic media can identify some microorganisms, and thus, we decided to investigate metagenomics data, which is a great approach for non-cultivable species [46]. The highest abundance was in the phylum *Bacteroidetes*, followed by *Firmicutes* and *Proteobacteria*. These phyla are major constituents of the gut microbiota of other birds and livestock animals [47]. *Salmonella* infections correlate with an increase in *Proteobacteria*, which agrees with our results from birds fed with no FRSM and URSM [48]. In turn, the *Firmicutes* phylum has previously been shown to increase after probiotic supplementation in broiler chickens [49]; our addition of *B. subtilis* strains (67 or 87Y) to RSM did not significantly change or even lower the abundance of this phylum.

The *Alistipes* genus and more specifically *Alistipes finegoldii* showed the highest abundance in the broilers’ ceca. *Alistipes* is generally considered a beneficial bacteria to the host gut [50]. It slightly increased and significantly decreased after supplementation with *B. subtilis* strains 87Y and 67, respectively. *Blastocystis* is an opportunistic pathogen and has previously been shown to cause a wide range of gastrointestinal tract symptoms [51]. Importantly, the addition of fermented or unfermented rapeseed meal showed inhibition of protozoan *B. hominis* development.

Alterations to the intestinal mucosa have huge impacts on the digestibility and growth performance of animals. We found no changes in the growth of the broilers during the experiment (Appendix A), and we expected that the cecal epithelium in the tested birds would be unchanged. Some authors have observed an improvement in the intestinal epithelial morphology of chickens fed various feed materials. Fermented feed materials seem to have a greater impact on intestinal morphology than raw materials. For example, a feed additive of 10% and 20% cottonseed meal fermented by *B. subtilis, Aspergillus oryzae*, and *A. niger* [52] or wheat bran fermented with *Trichoderma pseudokoningii* [53] increased the height of the villi, as well as the villi height/crypt depth ratio in the intestines of broiler chickens. Treatment of corn–soybean meal with a 10% addition of rapeseed meal fermented with *L. fermentum, E. faecium, S. cerevisiae,* and *B. subtilis* [54] and almost a 25% addition of fermented rapeseed meal by *B. subtilis, C. utilis,* and *E. faecalis* had similar positive effects on the duodenal, jejunum, ileum villi, and crypts in broilers [54].

Deformations in the negative control group (birds fed with no addition) may result in impaired absorption and reduced functionality of individual villi. The unfavorable representation of NC cecal morphology may be a result of the presence of *Salmonella* in the experimental group. These results agree with Borsoi et al., who showed that infections with pathogens such as *Salmonella* reduce the villi height/crypt depth ratio due to possible damage to the intestinal mucosa in broiler chickens [55]. In turn, the improvement of intestinal morphology makes nutrients more available for absorption [56,57,58]. The host microbiota and intestinal epithelium act as natural barriers to the movement of pathogenic bacteria, antigens, and toxins inside the gut mucosa; therefore, improvements in the gut morphology, such as the villi height and crypt depth, might be associated with an increased number of beneficial bacteria within the gut microbiome [59]. Hence, visible goblet cells in the mucosa of birds receiving *B. subtilis* 87Y FRSM are a desirable outcome. Simultaneously with a slight increase in bacteria in groups 67 and 87Y, the feed with 3% addition of fermented *B. subtilis* RSM caused noticeable changes in the species composition of the microorganisms living in the ceca of fed chickens.

Chicken supplementation with fermented or unfermented RSM did not significantly affect such cecal morphometric indices as height, width, or the distance between villi and the number of villi. However, in these groups of chickens, no deformities of the cecal mucosa were observed, which were found in the cecal epithelium of birds not receiving the feed supplement. Moreover, the morphological representation of the cecal specimens showed symptoms that may indicate increased immunological activity in the ceca of chickens not fed with RSM. These results concur with the results of pathogen detection (*Salmonella* spp., *K. pneumoniae*, and *E. cloacae* detected in the NC group), which supports the protective effects of FRMS on the gastrointestinal tract of broilers.

## 5. Conclusions

This study aimed to investigate the changes in the microbial composition of the cecum and histomorphometric analysis of its epithelium in broilers fed with feed mixture containing 3% of URSM or *B. subtilis* FRSM. Limitations in the use of FRSM relate to cost-effectiveness. The use of an FRSM content higher than 5% in compound feed is economically unjustified. Higher doses increase the feed mixture’s price; hence, future research should also focus on reducing the cost of the fermentation of rapeseed meal. The addition of FRSM stimulated the growth of some probiotic bacteria (*L. crispatus/johnsonii, L. reuterii*) and inhibited the growth of some pathogenic bacteria (*E. cloacae*, *K. pneumoniae*) and pathogenic *Eukaryote* (*Blastocyctis hominis*). The obtained results indicate that URSM also benefited microbial populations in the chicken cecum by stimulating the abundance of probiotic *L. salivarus* and inhibiting pathogenic *K. pneumoniae* and *B. hominis*. In addition, the histological structure of the cecum of chickens fed with FRSM or URSM was improved, indicating possible postbiotic and potential probiotic properties of this supplement. Future research directions in this area should include the development of a microbiological processing method of post-extraction rapeseed meal into FSRM, providing the possibility to obtain symbiotic effects and, consequently, the higher health status of the digestive tract of chickens; limited use of antimicrobials; higher bioavailability of nutrients from feed due to the enzymatic activity of fermentation bacteria and lower content of anti-nutritional substances, enabling higher doses of rapeseed meal.

## Figures and Tables

**Figure 1 microorganisms-09-00360-f001:**
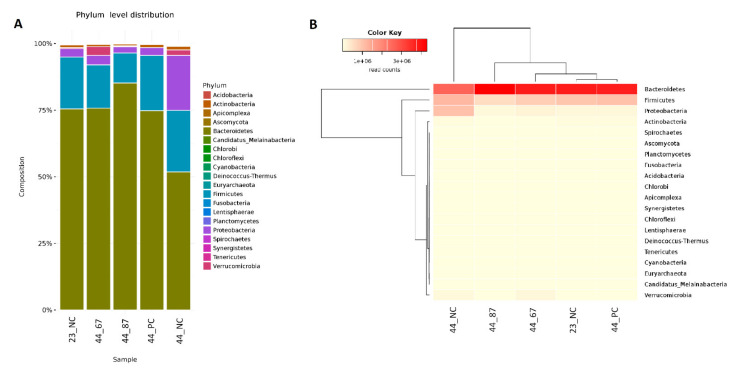
(**A**) The abundance of phyla and their relationships across the samples. (**B**) Dendrograms determined by computing hierarchical clustering from the abundance levels, showing the relationship between the phyla (left) and the samples (top). The abundance levels (number of reads associated with each taxa) were logarithmically transformed to base 2 for clarity.

**Figure 2 microorganisms-09-00360-f002:**
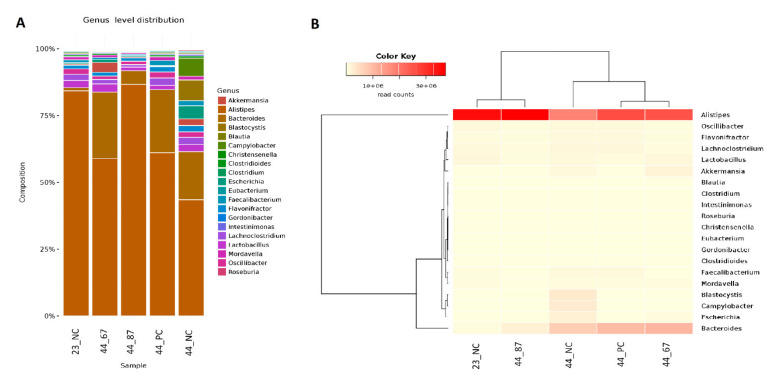
(**A**) The abundance of the genera and their relationships across the samples. (**B**) Dendrograms determined by computing hierarchical clustering from the abundance levels, showing the relationship between the genera (left) and the samples (top). The abundance levels (number of reads associated with each taxa) were logarithmically transformed to base 2 for clarity.

**Figure 3 microorganisms-09-00360-f003:**
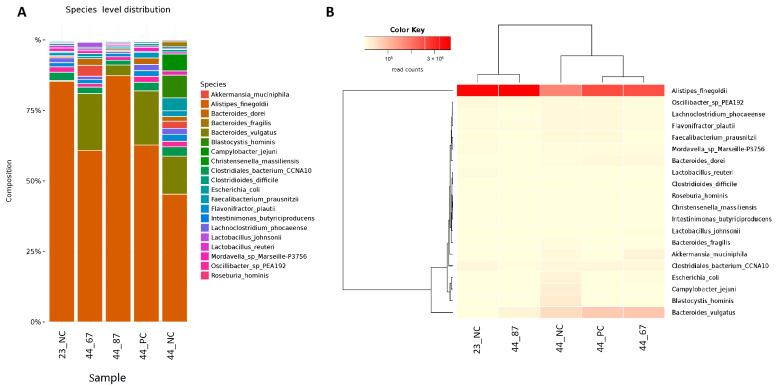
(**A**) Species abundance and their relationships across samples. (**B**) Dendrograms determined by computing hierarchical clustering from the abundance levels, showing the relationship between the species (left) and the samples (top). The abundance levels (number of reads associated with each taxa) were logarithmically transformed to base 2 for clarity.

**Figure 4 microorganisms-09-00360-f004:**
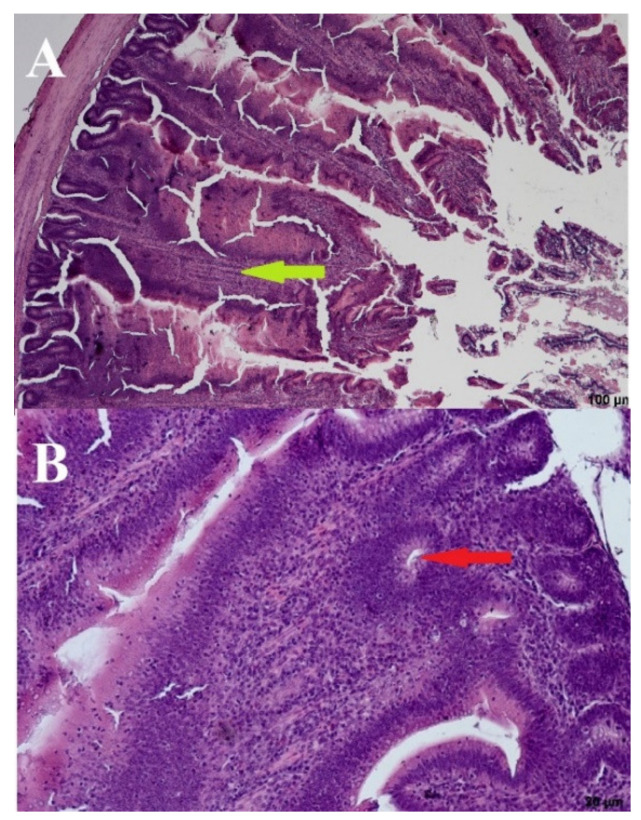
Negative control—(**A**) Cross section through the intestine—100 µm and (**B**) crypts—20 µm. Arrows: bold green–villi, red–crypts.

**Figure 5 microorganisms-09-00360-f005:**
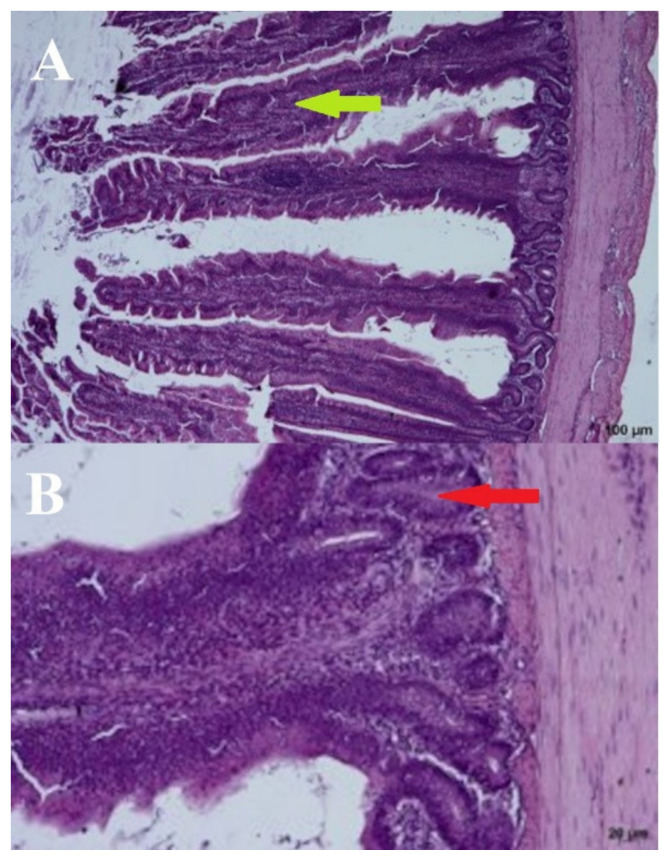
Positive control—(**A**) Cross section through the intestine—100 µm and (**B**) crypts–20 µm. Arrows: bold green—villi, red—crypts.

**Figure 6 microorganisms-09-00360-f006:**
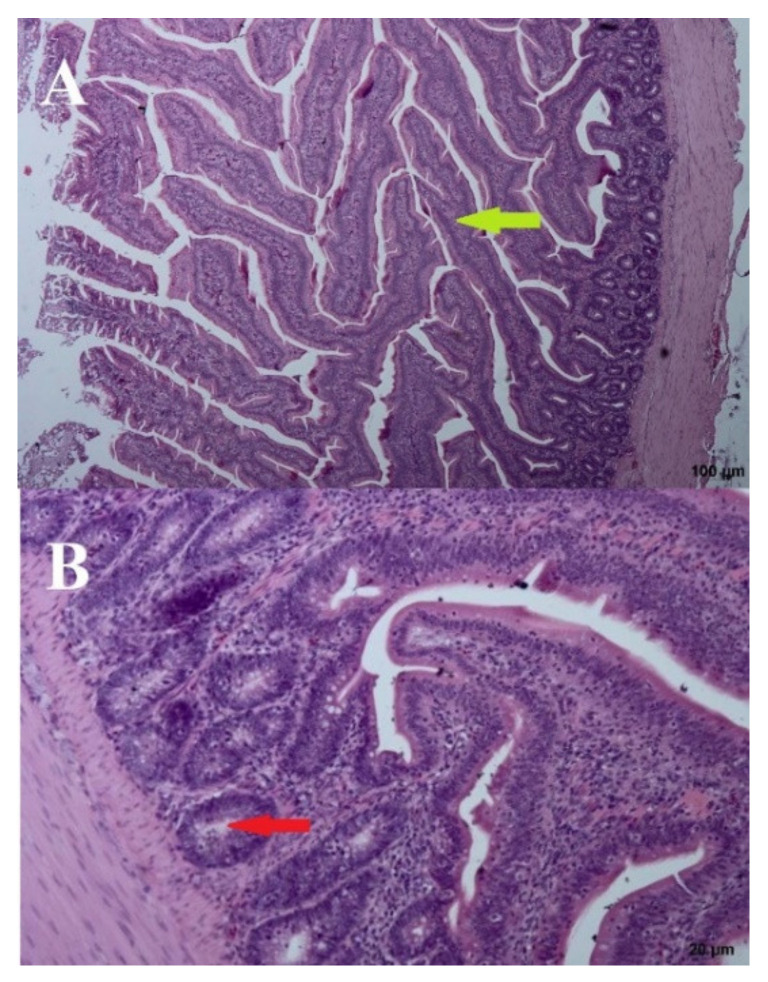
*B. subtilis* 67—(**A**) Cross section through the intestine—100 µm and (**B**) crypts—20 µm. Arrows: bold green—villi, red—crypts.

**Figure 7 microorganisms-09-00360-f007:**
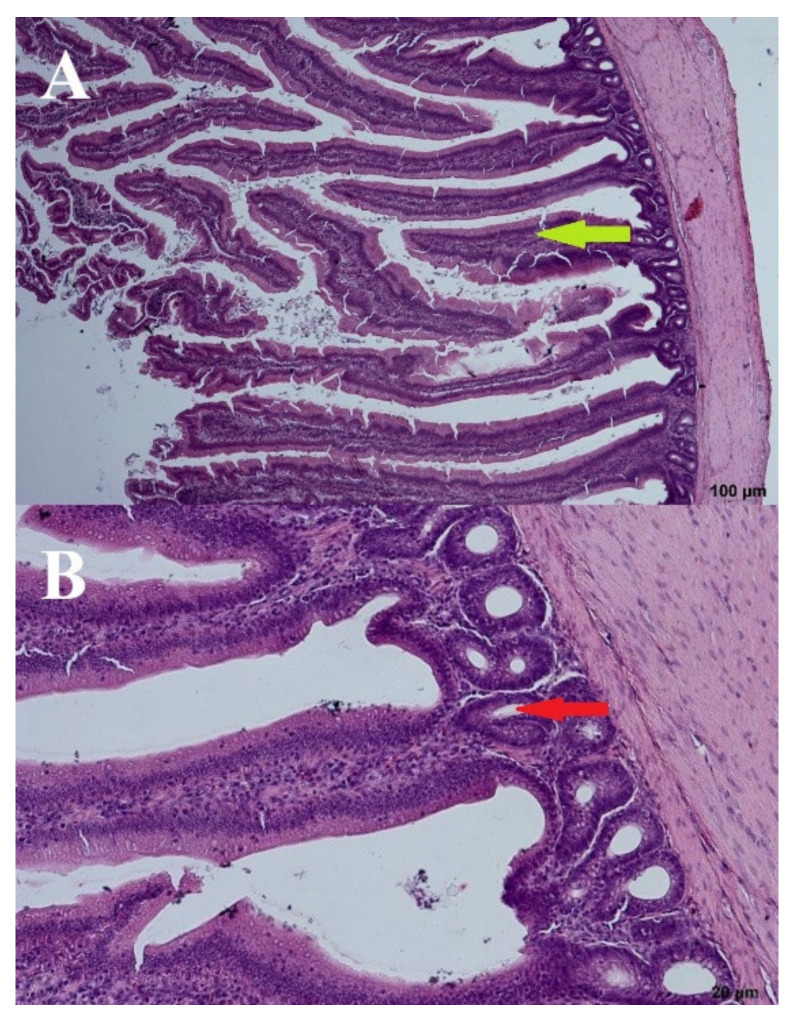
*B. subtilis* 87Y—(**A**). Cross section through the intestine—100 µm and (**B**). crypts—20 µm. Arrows: bold green—villi, red—crypts.

**Table 1 microorganisms-09-00360-t001:** Composition of the basal diet.

Component	Amount (kg)
Maize	361.87
Wheat	300.00
Soybean meal 45	217.00
Animal fat	30.00
Sunflower seed cake	25.00
Sunflower meal	15.00
Medium chain fatty acids	12.00
Guar 60	10.00
Chalk	7.30
Dicalcium phosphate	7.00
L-methionine 99	2.40
Salt	2.10
L-lysine SO4	2.00
Premix	2.00
L-lysine 98	1.70
Sodium bicarbonate (NAHCO3)	1.20
L-Threonine	1.10
Choline chloride 75%	0.67
Sacox 120/Kokcisan/Salinomax	0.58
Mycofix select 5.E	0.50
L-valine	0.44
Hiphos Liquid (L) (phytase)	0.08
Hostazym (xylanase)	0.06

**Table 2 microorganisms-09-00360-t002:** MALDI-TOF-MS identification of the bacteria species found on RAPID *Salmonella* agar. Range of score values: 2.3–3.0: Highly probable species identification; 2.0–2.29: Secure genus identification and probable species identification; 1.7–1.99: Probable genus identification; 0.0–1.69: Non-reliable identification.

Colony Color	MALDI Identification	Score Value
Violet	*Salmonella* spp.	2.3–3.0
Blue	*Klebsiella pneumoniae*	2.3–3.0
Green	*Enterobacter cloacae*	2.3–3.0
White	*Escherichia coli*	2.3–3.0

**Table 3 microorganisms-09-00360-t003:** Detection of the microorganisms found within amplified cecal material using RAPID *Salmonella* agar. NC, negative control; PC, birds receiving 3% addition of unfermented rapeseed meal (URSM); 67, birds receiving 3% addition of FRSM by *B. subtilis* 67; 87Y, birds receiving 3% addition of FRSM by *B. subtilis* 87Y. Species names defined by MALDI-TOF-MS.

Day of Experiment	Group	*Salmonella* spp.	*Klebsiella* *pneumoniae*	*Enterobacter* *cloacae*	*Escherichia coli*
21	NC	¼	0/4	0/4	¾
44	NC	0/4	2/4	¾	¾
44	PC	0/4	0/4	¾	¾
44	67	0/4	0/4	0/4	4/4
44	87Y	0/4	0/4	0/4	4/4

**Table 4 microorganisms-09-00360-t004:** Quantitative analysis of broiler cecal microbiome using chromogenic agar media. NC, negative control; PC, birds receiving 3% addition of URSM; 67, birds receiving 3% addition of FRSM by *B. subtilis* 67; 87Y, birds receiving 3% addition of FRSM by *B. subtilis* 87Y. Species names defined by MALDI-TOF-MS. RS, RAPID *Salmonella* agar; SB, StrepB select agar; MRS, MRS agar medium.

Microorganisms (log_10_ cfu/mg of Feces)	Start Day 21	NC Day 44	PC Day 44	67 Day 44	87Y Day 44	SEM	*p*-Value
*Escherichia coli* (RS)	2.90 ^Aa^	3.47 ^b^	3.06 ^AaBb^	3.56 ^B^	3.54 ^B^	0.06	0.000754
*Enterococcus faecalis* (SB)	1.23	0.90	1.99	1.00	2.58	0.22	0.0812
*Lactobacillus gasseri* (SB)	4.36	3.73	4.13	2.65	3.20	0.19	0.0574
*Streptococcus alactolyticus* (SB)	1.06	2.33	1.11	2.14	1.50	0.24	0.439
*Lactobacillus salivarus* (SB)	4.94 ^A^	2.94 ^B^	3.81 ^B^	n.d.	2.58 ^B^	0.26	0.00001
*Enterococcus faecium* (SB)	1.94	1.45	2.98	n.d.	1.25	0.23	0.175
*Lactobacillus crispatus/johnsonii* (MRS)	2.44 ^A^	2.61 ^A^	3.02 ^AB^	0.76 ^A^	4.41 ^B^	0.26	0.00001
*Lactobacillus reuterii* (MRS)	5.19 ^a^	4.19 ^A^	3.61 ^Ab^	5.53 ^B^	n.d.	0.28	0.00001
*Lactobacillus salivarus* (MRS)	4.21 ^A^	2.40 ^B^	4.24 ^A^	3.62 ^B^	3.58 ^B^	0.16	0.0047

^a,b^ Statistically significant differences between the groups for individual parameters, *p* < 0.05. ^A,B^ Statistically significant differences between the groups for individual parameters, *p* < 0.01.

**Table 5 microorganisms-09-00360-t005:** MALDI-TOF-MS identification of bacteria species found on StrepB select agar and MRS (DeMan, Rogosa, and Sharpe) agar. Range of score values: 2.3–3.0: Highly probable species identification; 2.0–2.29: Secure genus identification and probable species identification; 1.7–1.99: Probable genus identification; 0.0–1.69: Non-reliable identification.

Chromogenic Medium	Colony Color	MALDI Identification	Score Value
StrepB agar	Blue	*Enterococcus faecalis*	2.3–3.0
	Lilac	*Enterococcus faecium*	2.3–3.0
	Violet matte	*Lactobacillus gasseri*	2.3–3.0
	Light violet	*Streptococcus alactolyticus*	1.7–1.99
	Violet/slight pink	*Lactobacillus salivarus*	2.0–2.29
MRS	Large white	*Lactobacillus salivarus*	1.7–1.99
	Matt white	*Lactobacillus crispatus/johnsonii*	2.3–3.0
	Transparent	*Lactobacillus reuterii*	2.0–2.29

**Table 6 microorganisms-09-00360-t006:** Taxonomic profiling metrics.

Sample Name	Reads	Classified	Unclassified
**21_NC**	66,689,620	14,624,156 (21.93%)	52,065,464 (78.07%)
**44_NC**	59,653,504	11,070,866 (18.56%)	48,582,638 (81.44%)
**44_PC**	52,907,882	13,069,908 (24.7%)	39,837,974 (75.30%)
**44_67**	62,205,312	17,429,830 (28.02%)	44,775,482 (71.98%)
**44_87Y**	61,418,084	21,317,558 (34.71%)	40,100,526 (65.29%)

**Table 7 microorganisms-09-00360-t007:** Number of reads assigned to different kingdoms. Ambiguous—reads which cannot be assigned to one specific kingdom; *Eukaryote*—parasitic and non-parasitic *Protozoa*.

Kingdom	21_NC	44_NC	44_PC	44_67	44_87Y
Archaea	3436 (0.02%)	25,676 (0.23%)	3714 (0.03%)	4010 (0.02%)	4114 (0.02%)
Bacteria	14,551,854 (99.51%)	10,301,176 (93.06%)	13,015,864 (99.59%)	17,363,714 (99.62%)	21,262,068 (99.74%)
Eukaryote	19,684 (0.13%)	663,974 (6.0%)	6522 (0.05%)	6582 (0.04%)	6450 (0.03%)
Fungi	6252 (0.04%)	8746 (0.08%)	4526 (0.03%)	5142 (0.03%)	4822 (0.02%)
Viruses	2622 (0.02%)	4970 (0.04%)	5460 (0.04%)	9946 (0.06%)	1010 (0.00%)
Ambiguous	40,308 (0.28%)	65,324 (0.59%)	33,822 (0.26%)	40,440 (0.23%)	39,094 (0.18%)

## Data Availability

The data presented in this study are available on request from the corresponding author.

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
