# Peer review of "Changes in the Microbial Composition of the Cecum and Histomorphometric Analysis of Its Epithelium in Broilers Fed with Feed Mixture Containing Fermented Rapeseed Meal"

_microorganisms, 2021, doi:10.3390/microorganisms9020360_

Round 1

Reviewer 1 Report

Lines 188-189 “Groups fed with  fermented rapeseed meal were not infected by E. cloacae (Table 3).” This is not true. Please change

Lines 192-194 “The number of E. coli found in raw material from the first to the last  day of the experiment doubled in the negative control group (NC). At day 44, the abundance of E. coli was slightly different in groups fed with additives (PC, 67, 87Y) (Table 3)” It is not possible to verify that affirmed from Table 3. I suggest to delete the Table 3

Lines 217-219 “Only broilers receiving 3% of B. subtilis 67 FRSM (d: 44, g: 67)  had a reduction in this pathogen at the end of the experiment (Table 5). In turn, the abundance of E.  faecium largely decreased after 44 days independent of treatment” Please, delete: the differences are not statistically significant

Line 229 change (Table 6) as (Table 5).

Lines 226-231 “The last colored bacterial morphology found on the strepB agar was Streptococcus alactolyticus  (Table 4). Populations of S. alactolyticusfound in the chicken cecum decreased during the study period  in the negative control (NC) and positive control (PC) as well as within chickens receiving 3% of  FRSM by B. subtilis 87Y (87Y) (Table 6). Unexpectedly, the addition of 3% of RSM fermented by B.  subtilis 67 (d: 44, g: 67) caused a considerable increase of S. alactolyticus at the end of the experiment  (Table 5).” The results are not in line with the values in Table 5. Please check the period.

Line 369 – after pathogenic infections, please add the following reference (Zou, X.; Xiao, R.; Li, H.; Liu, T.; Liao, Y.; Wang, Y,; Wu, S.; Li, Z.  Effect of a novel strain of Lactobacillus brevis M8 and tea polyphenol diets on performance, meat quality and intestinal microbiota in broilers. Ital. J. Anim. Sci. 2018, 17, 396–407)

Author Response

Thank you for your comments and suggestions that have helped us improve our work. Our answers are presented below, and the corrections are highlighted in the re-attached manuscript.

Lines 188-189 “Groups fed with  fermented rapeseed meal were not infected by E. cloacae (Table 3).” This is not true. Please change

Current lines: 193-196. E. cloacae were not detected in both groups fed with fermented rapeseed meal (67 and 87Y, Table 3). Group PC infected by E. cloacae  received 3% of UNfermented rapeseed meal.

Lines 192-194 “The number of E. coli found in raw material from the first to the last  day of the experiment doubled in the negative control group (NC). At day 44, the abundance of E. coli was slightly different in groups fed with additives (PC, 67, 87Y) (Table 3)” It is not possible to verify that affirmed from Table 3. I suggest to delete the Table 3

We apologize for the mistake, it was Table 5. It is corrected in line 201.

Lines 217-219 “Only broilers receiving 3% of B. subtilis 67 FRSM (d: 44, g: 67)  had a reduction in this pathogen at the end of the experiment (Table 5). In turn, the abundance of E.  faecium largely decreased after 44 days independent of treatment” Please, delete: the differences are not statistically significant

Thank you for your suggestion. Line 226-228 have been deleted. 

Line 229 change (Table 6) as (Table 5).

Lines 226-231 “The last colored bacterial morphology found on the strepB agar was Streptococcus alactolyticus  (Table 4). Populations of S. alactolyticus found in the chicken cecum decreased during the study period  in the negative control (NC) and positive control (PC) as well as within chickens receiving 3% of  FRSM by B. subtilis 87Y (87Y) (Table 6). Unexpectedly, the addition of 3% of RSM fermented by B.  subtilis 67 (d: 44, g: 67) caused a considerable increase of S. alactolyticus at the end of the experiment  (Table 5).” The results are not in line with the values in Table 5. Please check the period.

As noted by the reviewer, changes in population of Streptococcus alactolyticus were not statistically significant so we deleted these lines 236-241.

Line 369 – after pathogenic infections, please add the following reference (Zou, X.; Xiao, R.; Li, H.; Liu, T.; Liao, Y.; Wang, Y,; Wu, S.; Li, Z.  Effect of a novel strain of Lactobacillus brevis M8 and tea polyphenol diets on performance, meat quality and intestinal microbiota in broilers. Ital. J. Anim. Sci. 2018, 17, 396–407)

Thank you for suggestion. We added this reference as position 42.

Reviewer 2 Report

Interesting paper by Szmigiel et al. on fermented rapeseed as a carrier of a potential probiotic in broilers, but the manuscript has several drawbacks which deny publication.

Line 27 …addition of 3% FRSM has prebiotic and probiotic effect…on what basis can you give such a conclusion…there was no experimental group only receiving fermented rapeseed meal…this is pure speculation. Especially as in fermented RSM easily fermentable carbohydrates (prebiotics) have been degraded. If it all you may have a possible positive effect by degradation of antinutritive factors or production of organic acids during fermentation…but these are no prebiotic effects!!!

Line 42…why would you call rapeseed meal protein a high quality protein , it is low in lysine!!

Line 50…Reference 13, 14 are about FRSM in sows does not fit here for broiler experiments. There are significant differences in the morphology and function of the digestive tract between broiler and sows which deny comparison …please delete.

Line 51..what do you mean with high value added product…please be specific …what is the higher value???

Line 56…limited research on effects of FRSM on growth results…??? In the manuscript no growth results are presented!!

Line 58-59… why was only the microbial composition of the ceca  investigated…please give a reason!!! …morphology of the gastrointestinal tract is too much ….gastrointestinal tract starts with the beak and ends with the cloaca in birds and you only investigated the ceca!!

General statement: in this experiment two effects are investigated 1) changes in FRSm (destruction of antinutritive factors, production of lactic  acid,…) and 2) possible probiotic effects of live Bacillus subtilis (spores or bacterium??? So it is unclear if there was a probiotc, a feed  or  a synergistic effect of both. However, to show the effect of only FRSM  an experimental group with 3 % of autoclaved FRSM should have been included in the experiment .

Line 61-69…What is the bacteria concentration in the inoculum and as well what is the bacteria concentration in the FRSM. Without these informations it is not possible to repeat this study or to compare it with other studies as probiotic effects are dependent on the dose!!

Line 72 please include standard deviation of the live weight.

Line 74 give a reference for breeders recommendations

Line 76… 10 minutes light provided every 6 hours??? Is this correct? Is this conform with animal rights

Line 84 here you claim rapeseed meal was used as an additive?? What does this mean. I guess you mean carrier of your potential probiotic. FRSM is still a feedstuff and not an additive…please reword!

Line 85..Firstly, why was 3 % FRSM added and what is the corresponding number of bacteria dose in the 3% FRSM. Secondly how do you know that 3 % did not change the content of feed mixture. There is no clear description on which expense the 3% FRSM was included in the basal diet, and furthermore no analysis of the feedstuffs have been performed , so how do you know that there is no difference between each of the diets with respect to nutrient content, ANF etc.

Line 87-89..Was feed intake in all experimental groups the same, this may as well affect microbial composition or at least total numbers.

Line 92..has there been any approvement of this study by an ethics committee or was this study with animal care …please indicate

Line 96.. what do you mean with cecum of each bird was removed? Usually birds have two ceca, did you remove both and did you pool samples from both ceca…please describe precisely. Is it possible that at different sites of the ceca the microbial composition may be different???

Line 124 please do not use rpm use the centrifugal force

Please check supplier details (company, City, Country) in many cases only the company has been given…check as well abbreviations (line 125 HCCA?...)

Line 131 …total intestinal contents??? You only took samples from the ceca!!!

Line 143 from which ceca and from which site of the ceca…more precision

Line 160-161…this was described in the Materail and Method Section …can be deleted here

Line 175 Sentence sounds not correct!!

Line 193-194…This cannot be seen in Table 3. ??? Were did you get these results from???

Line 218-219 …there was no significant effect, so there is no reduction in E. faecium and faecalis…please rewrite or delete

Line 230 again , presentation of non significant results…there is no increase when not significant different

Line 286-286 again presentation of non signifcant results…there is no increase when not statistically significant

295-320 nice pictures but no quantitative traits, quantitative traits have been shown not to be different.

Line 312 this belongs in the discussion. Not only number of goblet cells is of importance, size of the cells as well. Did you calculate the number of goblet cells then present these here. What kind of mucin will be secreted by goblet cells is as well of importance

Line 324..has not been shown that feed mixture was not changed???

Line 360 These birds had no Salmonella in their microbiome, so why do authors conclude that 3 % FRSm with B subtilis has bacteriostatic effects against pathogens???

Line 375-378 Authors insist that they didn#T change the feed mixture or nutrient content, but now blame the lactic acid to control pathogens…so there was a change in feed mixture , but this has not been shown…please analyse the feeds…all diets for crude nutrients, fermentation metabolites and antinutritive factors without this the manuscript is without any use and only speculative. And no comparison to earlier or later studies can be done.

Line 397…what have herbal extracts to do with FRSM and possible probiotics effects…

Lines 407-420 The discussion should explain why you didn’t find any difference instead you declare your non significant results to be different…needs to be rewritten

Line 421…increased lymphocyte count were does this now come from…there is no table showing quantitative lymphocyte counts.

Interesting paper by Szmigiel et al. on fermented rapeseed as a carrier of a potential probiotic in broilers, but the manuscript has several drawbacks which deny publication.

Line 27 …addition of 3% FRSM has prebiotic and probiotic effect…on what basis can you give such a conclusion…there was no experimental group only receiving fermented rapeseed meal…this is pure speculation. Especially as in fermented RSM easily fermentable carbohydrates (prebiotics) have been degraded. If it all you may have a possible positive effect by degradation of antinutritive factors or production of organic acids during fermentation…but these are no prebiotic effects!!!

Line 42…why would you call rapeseed meal protein a high quality protein , it is low in lysine!!

Line 50…Reference 13, 14 are about FRSM in sows does not fit here for broiler experiments. There are significant differences in the morphology and function of the digestive tract between broiler and sows which deny comparison …please delete.

Line 51..what do you mean with high value added product…please be specific …what is the higher value???

Line 56…limited research on effects of FRSM on growth results…??? In the manuscript no growth results are presented!!

Line 58-59… why was only the microbial composition of the ceca  investigated…please give a reason!!! …morphology of the gastrointestinal tract is too much ….gastrointestinal tract starts with the beak and ends with the cloaca in birds and you only investigated the ceca!!

General statement: in this experiment two effects are investigated 1) changes in FRSm (destruction of antinutritive factors, production of lactic  acid,…) and 2) possible probiotic effects of live Bacillus subtilis (spores or bacterium??? So it is unclear if there was a probiotc, a feed  or  a synergistic effect of both. However, to show the effect of only FRSM  an experimental group with 3 % of autoclaved FRSM should have been included in the experiment .

Line 61-69…What is the bacteria concentration in the inoculum and as well what is the bacteria concentration in the FRSM. Without these informations it is not possible to repeat this study or to compare it with other studies as probiotic effects are dependent on the dose!!

Line 72 please include standard deviation of the live weight.

Line 74 give a reference for breeders recommendations

Line 76… 10 minutes light provided every 6 hours??? Is this correct? Is this conform with animal rights

Line 84 here you claim rapeseed meal was used as an additive?? What does this mean. I guess you mean carrier of your potential probiotic. FRSM is still a feedstuff and not an additive…please reword!

Line 85..Firstly, why was 3 % FRSM added and what is the corresponding number of bacteria dose in the 3% FRSM. Secondly how do you know that 3 % did not change the content of feed mixture. There is no clear description on which expense the 3% FRSM was included in the basal diet, and furthermore no analysis of the feedstuffs have been performed , so how do you know that there is no difference between each of the diets with respect to nutrient content, ANF etc.

Line 87-89..Was feed intake in all experimental groups the same, this may as well affect microbial composition or at least total numbers.

Line 92..has there been any approvement of this study by an ethics committee or was this study with animal care …please indicate

Line 96.. what do you mean with cecum of each bird was removed? Usually birds have two ceca, did you remove both and did you pool samples from both ceca…please describe precisely. Is it possible that at different sites of the ceca the microbial composition may be different???

Line 124 please do not use rpm use the centrifugal force

Please check supplier details (company, City, Country) in many cases only the company has been given…check as well abbreviations (line 125 HCCA?...)

Line 131 …total intestinal contents??? You only took samples from the ceca!!!

Line 143 from which ceca and from which site of the ceca…more precision

Line 160-161…this was described in the Materail and Method Section …can be deleted here

Line 175 Sentence sounds not correct!!

Line 193-194…This cannot be seen in Table 3. ??? Were did you get these results from???

Line 218-219 …there was no significant effect, so there is no reduction in E. faecium and faecalis…please rewrite or delete

Line 230 again , presentation of non significant results…there is no increase when not significant different

Line 286-286 again presentation of non signifcant results…there is no increase when not statistically significant

295-320 nice pictures but no quantitative traits, quantitative traits have been shown not to be different.

Line 312 this belongs in the discussion. Not only number of goblet cells is of importance, size of the cells as well. Did you calculate the number of goblet cells then present these here. What kind of mucin will be secreted by goblet cells is as well of importance

Line 324..has not been shown that feed mixture was not changed???

Line 360 These birds had no Salmonella in their microbiome, so why do authors conclude that 3 % FRSm with B subtilis has bacteriostatic effects against pathogens???

Line 375-378 Authors insist that they didn#T change the feed mixture or nutrient content, but now blame the lactic acid to control pathogens…so there was a change in feed mixture , but this has not been shown…please analyse the feeds…all diets for crude nutrients, fermentation metabolites and antinutritive factors without this the manuscript is without any use and only speculative. And no comparison to earlier or later studies can be done.

Line 397…what have herbal extracts to do with FRSM and possible probiotics effects…

Lines 407-420 The discussion should explain why you didn’t find any difference instead you declare your non significant results to be different…needs to be rewritten

Line 421…increased lymphocyte count were does this now come from…there is no table showing quantitative lymphocyte counts.

Author Response

Thank you for your comments and suggestions that have helped us improve our work. Our answers are presented below, and the corrections are highlighted in the re-attached manuscript.

Interesting paper by Szmigiel et al. on fermented rapeseed as a carrier of a potential probiotic in broilers, but the manuscript has several drawbacks which deny publication.

Line 27 …addition of 3% FRSM has prebiotic and probiotic effect…on what basis can you give such a conclusion…there was no experimental group only receiving fermented rapeseed meal…this is pure speculation. Especially as in fermented RSM easily fermentable carbohydrates (prebiotics) have been degraded. If it all you may have a possible positive effect by degradation of antinutritive factors or production of organic acids during fermentation…but these are no prebiotic effects!!!

Thank you for your suggestion. We actually speculate that B. subtilis 87Y and 67 strains are probiotic based mainly on our results published in Biomass Conv. Biorefinery 2019. Therefore we decided to write „potential probiotic” in our manuscript.

Line 42…why would you call rapeseed meal protein a high quality protein , it is low in lysine

Thank you for your suggestion. It is true that rapeseed meal has lower abundance of lysine than soybean meal. However, rapeseed meal has higher amount of sulfur aminoacids (methionine, cysteine and tryptophan), that are more important in poultry feeding. We considered nutritional value of rapeseed meal only in terms of poultry feeding. We have refined the information in our manuscript (line 42).

Line 50…Reference 13, 14 are about FRSM in sows does not fit here for broiler experiments. There are significant differences in the morphology and function of the digestive tract between broiler and sows which deny comparison …please delete

Thank you for your suggestion. We did delete these references (Grela et a. 2019. And Tomaszewska et al. 2019) and we cited other publications (line 49).

Line 51..what do you mean with high value added product…please be specific …what is the higher value???

Thank you. We expanded that paragraph in lines 48-49.

Line 56…limited research on effects of FRSM on growth results…??? In the manuscript no growth results are presented!!

 Thank you for your suggestion. We have rewrote the lines 56-58.

Line 58-59… why was only the microbial composition of the ceca  investigated…please give a reason!!! …morphology of the gastrointestinal tract is too much ….gastrointestinal tract starts with the beak and ends with the cloaca in birds and you only investigated the ceca

We have specified the place of our examination to the cecum and we rewrote mentioned sentence in lines 55-57.  At the beginning of the introduction (lines 36-39) we wrote why the cecum is important for investigations.

General statement: in this experiment two effects are investigated 1) changes in FRSm (destruction of antinutritive factors, production of lactic  acid,…) and 2) possible probiotic effects of live Bacillus subtilis (spores or bacterium??? So it is unclear if there was a probiotc, a feed  or  a synergistic effect of both. However, to show the effect of only FRSM  an experimental group with 3 % of autoclaved FRSM should have been included in the experiment .

We do agree with the reviewer that it would be more simple to study one parameter. However, it is impossible to modify one parameter (ex. Bacillus living cells) without changing other parameters. Autoclaving, besides destroying living cells, also modify chemical composition ex. Maillard reaction or decomposition of temperature-sensitive chemical compounds, protein digestibility due to its denaturation, and enzymes' inactivation. Thus, autoclaving does not enable to change just one parameter. In our previous study, we have shown potential probiotic activity of the used B. subtilis strain. This study aimed to show the possible probiotic activity in the natural environment of fermented RSM and possible interactions, synergisms with other parameters.

Line 61-69…What is the bacteria concentration in the inoculum and as well what is the bacteria concentration in the FRSM. Without these informations it is not possible to repeat this study or to compare it with other studies as probiotic effects are dependent on the dose!!

In inoculum at optical density OD = 0.1 there is 0.35 × 107 cfu / mL. After fermentation, the amount of bacteria is 1 × 107 cfu / mL. We added these informations in lines 63 and 68.

Line 72 please include standard deviation of the live weight.

SD of the live weight is 4,242. We added this data in line 72.

Line 74 give a reference for breeders recommendations

Thank you. The recommendations for breeders are available at: http://en.aviagen.com/assets/Tech_Center/Ross_Broiler/Ross-BroilerHandbook2018-EN.pdf

We added the reference in line 76.

Line 76… 10 minutes light provided every 6 hours??? Is this correct? Is this conform with animal rights

Thank you for pointing out our mistake. We did rewrote this sentence in lines 74-76.

Line 84 here you claim rapeseed meal was used as an additive?? What does this mean. I guess you mean carrier of your potential probiotic. FRSM is still a feedstuff and not an additive…please reword!

Thank you for your suggestion. We have rewrote this sentence in lines 83-85.

Line 85. Firstly, why was 3 % FRSM added and what is the corresponding number of bacteria dose in the 3% FRSM. Secondly how do you know that 3 % did not change the content of feed mixture. There is no clear description on which expense the 3% FRSM was included in the basal diet, and furthermore no analysis of the feedstuffs have been performed, so how do you know that there is no difference between each of the diets with respect to nutrient content, ANF etc.

Nutritional practice and standards say that rapeseed meal should not constitute more than 5% of the feed mix, so we decided that 3% would be a safe value. We added appropriate references in line 83.

Line 87-89. Was feed intake in all experimental groups the same, this may as well affect microbial composition or at least total numbers.

The feed intake was monitored and in all groups and was in accordance with the line manufacturer's instructions, therefore studied birds had the same conditions for the same food intake. We added appropriate reference in line 89-90.

Line 92. has there been any approvement of this study by an ethics committee or was this study with animal care …please indicate

Polish law, particularly an Act of 15.01.2015 on the Protection of Animals Used for Scientific and Educational Purposes, specifies terms and conditions on the protection of animals used for scientific or educational purposes, including conditions when an Ethical Approval is required. Ethical Approval is not required for veterinary services within the scope of the Act of 18.12.2003 on animal treatment facilities, as well as agricultural activities, including rearing or breeding of animals, carried out in accordance with the provisions on the protection of animals; and activities that, in compliance with the veterinary medicine practice, do not cause pain, suffering, distress or permanent damage to the body of animals, to an extent equal to a needle stick, or more intense. In this research:

  • the animals were maintained in the standard production conditions,
  • the animals were not exposed to pain and suffering in any way,
  • no blood samples were taken,
  • sampling of the excreta was not harmful to birds in any way.

Thus, the experiment did not require Ethical Approval under the abovementioned applicable law.

Additionally, we provided Animal Welfare Certificate.

Line 96. what do you mean with cecum of each bird was removed? Usually birds have two ceca, did you remove both and did you pool samples from both ceca…please describe precisely. Is it possible that at different sites of the ceca the microbial composition may be different???

We apologize for the misword. Of course, birds have two ceca and we did remove both of them and we did pool the samples from both of ceca. We have rewrote this statement in lines  96-97.

Line 124 please do not use rpm use the centrifugal force

Thank you for your suggestion. It has been corrected in line 125.

Please check supplier details (company, City, Country) in many cases only the company has been given…check as well abbreviations (line 125 HCCA?...)

Thank you. It has been corrected in line 127, also the check supplier details were added in all manuscript.

Line 131 …total intestinal contents??? You only took samples from the ceca!!!

Thank you for your suggestion. It has been corrected in line 133-134.

Line 143 from which ceca and from which site of the ceca…more precision

Samples were taken from proximal regions both cecal deviations. We added the information on line 144.

Line 160-161…this was described in the Materail and Method Section …can be deleted here

Thank you. It has been deleted.

Line 175 Sentence sounds not correct!!

Thank you. It has been corrected in line 181-182.

Line 193-194…This cannot be seen in Table 3. ??? Were did you get these results from???

We apologize for the mistake. We rewrote this sentence to Table 5 in line 201.

Line 218-219 …there was no significant effect, so there is no reduction in E. faecium and faecalis…please rewrite or delete

Thank you for your suggestion. We did delete this sentence (226-228).

Line 230 again, presentation of non significant results…there is no increase when not significant different. Line 286-286 again presentation of non signifcant results…there is no increase when not statistically significant

Thank you. Lines 226 - 231 (now – 236-241 and 286 (now – 301-302) were deleted.

295-320 nice pictures but no quantitative traits, quantitative traits have been shown not to be different.

Histology photos were presented only for illustration purposes, not to indicate quantitative characteristics - these are included in the table S1. Moreover, in the part of the publication on the morphology of the cecum epithelium, we used the vocabulary used in the I See Inside (ISI) method, which is more dynamic way than the traditional linear measurements of villi and crypts. ISI is successfully used and described by other teams dealing with the histology of the digestive tract of chickens (Sanches at al., 2020; Belote et al., 2019; Belote et al., 2018). We have added relevant information and citations in the chapter M&M.

Line 312 this belongs in the discussion. Not only number of goblet cells is of importance, size of the cells as well. Did you calculate the number of goblet cells then present these here. What kind of mucin will be secreted by goblet cells is as well of importance

We did not quantify goblet cells or the type of mucin. Our histomorphological results only show a general picture of changes in cecum cells.

Line 324. has not been shown that feed mixture was not changed???

Thank you for your suggestion. This sentence was deleted, lines 338-339.

Line 360 These birds had no Salmonella in their microbiome, so why do authors conclude that 3 % FRSm with B subtilis has bacteriostatic effects against pathogens???

We do conclude that 3% of fermented rapeseed meal may have bacteriostatic effect because group NC –negative control receiving no additions, was infected by Salmonella. It is shown in Table 3. We changed the phrase „bacteriostatic effect” to „inhibits the development of pathogens”, which at the present stage of our research better illustrates the obtained results. Lines 376 and 377. Thanks for your suggestion.

Line 375-378 Authors insist that they didn#T change the feed mixture or nutrient content, but now blame the lactic acid to control pathogens…so there was a change in feed mixture , but this has not been shown…please analyse the feeds…all diets for crude nutrients, fermentation metabolites and antinutritive factors without this the manuscript is without any use and only speculative. And no comparison to earlier or later studies can be done.

Thank you for your suggestion. Lines 381-384 (now – 390-393 – highlighted in yellow) have been deleted.

Line 397…what have herbal extracts to do with FRSM and possible probiotics effects

Thank you. We have deleted this sentence, lines 412-414.

Lines 407-420 The discussion should explain why you didn’t find any difference instead you declare your non significant results to be different…needs to be rewritten …

Thank you for suggestion. We rewrote this part of the discussion (lines 435-440). We did not observe differences in parameters such as the length, width of the villi or the distance between them, but we noticed that in the cecum of chickens fed with rapeseed meal there was no deformation that existed in the cecal epithelium of the birds in the negative control. In the epithelium of the cecum of birds not fed with rapeseed meal, we observed symptoms of an increased immune response, which probably arose in response to pathogens identified by us in this part of the intestine.

Line 421…increased lymphocyte count were does this now come from…there is no table showing quantitative lymphocyte counts.

According to information obtained from histopathologists, in our research the lymphocyte count is not representative (the distribution on the mucosa is irregular). The appearance of pro-inflammatory cells indicates inflammation, and their number is defined descriptively, in acute conditions there are infiltrates in which the number of lymphocytes cannot be determined.

Round 2

Reviewer 2 Report

Review of the revised manuscript by Szmigiel et al on the effect of fermented rapeseed as carrier of a potential probiotic.

The title needs to be changed as the manuscript reads now only the potential probiotics are of interest, while the title indicates that  the main topic is fermented rapeseed meal.

Line 67…the authors indicate the B. subtilis concentration in cfu/ml. This is the bacteria number after fermentation. We need to know the amount of cfu/g or kg diet as fed. As diets were pelleted (line 87) the amount of cfu/ml of the fermented wet product is not representative of the feed.  In addition how was the fermentation stopped after 24 h…not indicated. Were there difference between B. subtilis 67 and 87Y with respect to cfcu/g or kg diet ?

Line 82-84 the formulation of the diet is not given, it is unclear on the expense of which other dietary component the rapeseed meal  unfermented or fermented with Bacillus was included in the diet.The addition of rapeseed meal unfermented or fermented leads possbly to increases or decreases in specific nutrients in the diet which may as well affect the cecal  microbiom. Therefore the nutrient composition should be as well analysed without this information it is unclear what is an effect of the probiotic and what is a dietary effect.

Line 88 …birds received feed…feed intake in accordance with the manufactures instructions?????  Please include the feed intake…This is science…we need numbers

Line 375.. I do not understand on what basis the authors come to the conclusion that 3 % FRSM with B subtilis inhibits development of pathogens…E coli increases, Salmonella were not existent in the cecum and enteroccoci and streptocooci were not changed.. looking closer at it, the inclusion of unfermented rape seed meal seems more effective against pathogens than the fermented.

Line 408 IF there are growth data , why don‘T you include them here this would give a more fully picture of the experiment, but to write there was no effect and data not shown is not acceptable..include growth data!!

Line413… fermented feed additives?? Fermented feed…feed additives have to be registered and usually are given in very low amounts to the diets. By the way as you may notice all these studies used higher amounts oft he respective fermented feed. I still haven’t found any explanation or rationale why only 3 % FRSM was added

Line 446 even the unfermented rapeseed stimulated L. salivarus and had numerical reduction of other bacteria so why do  you recommend only the fermented rapeseed meal...maybe good if you discuss as well the the results of the unfermented RSM

Line 447  … there were no Eukaryote measured in this study…your conclusion should be based on the results of present study and not previous or other peoples studies.

Author Response

Comments and Suggestions for Authors

Review of the revised manuscript by Szmigiel et al on the effect of fermented rapeseed as carrier of a potential probiotic.

The title needs to be changed as the manuscript reads now only the potential probiotics are of interest, while the title indicates that  the main topic is fermented rapeseed meal.

As suggested, we have changed the title of the manuscript

Line 67…the authors indicate the B. subtilis concentration in cfu/ml. This is the bacteria number after fermentation. We need to know the amount of cfu/g or kg diet as fed. As diets were pelleted (line 87) the amount of cfu/ml of the fermented wet product is not representative of the feed.  In addition how was the fermentation stopped after 24 h…not indicated. Were there difference between B. subtilis 67 and 87Y with respect to cfcu/g or kg diet ?

Fermentation after 24 hours was stopped by drying in the dryer. Fermented rapeseed meal was added to the feed mixture after drying. The number of colony forming units (CFU) was estimated using dried feed material, taking into account the decrease in CFU due to the heat treatment. Both used strains multiplied at the same rate and achieved titers between which we did not observe statistically significant differences. The final amount of B. subtilis cells in the fermented feed was 1.8x10^7 CFU / g. We have added this information in lines 66-67.

Line 82-84 the formulation of the diet is not given, it is unclear on the expense of which other dietary component the rapeseed meal  unfermented or fermented with Bacillus was included in the diet. The addition of rapeseed meal unfermented or fermented leads possbly to increases or decreases in specific nutrients in the diet which may as well affect the cecal  microbiom. Therefore the nutrient composition should be as well analysed without this information it is unclear what is an effect of the probiotic and what is a dietary effect.

The composition of the feed is given in table 1. Nutritional practice and standards say that rapeseed meal should not constitute more than 5% of the feed mix, so we decided that 3% would be a safe value. We added appropriate references in line 82.

In the control group of chickens, we used feed without a rapeseed meal or fermented rapeseed meal. All our experiments also included the results of the control group of birds. Hence the effect or no effect of the additive used can be determined by comparing all broilers' groups, what we did.

Line 88 …birds received feed…feed intake in accordance with the manufactures instructions?????  Please include the feed intake…This is science…we need numbers

We have added a table with the relevant data to the supplements.

Line 375.. I do not understand on what basis the authors come to the conclusion that 3% FRSM with B subtilis inhibits development of pathogens…E coli increases, Salmonella were not existent in the cecum and enteroccoci and streptocooci were not changed.. looking closer at it, the inclusion of unfermented rape seed meal seems more effective against pathogens than the fermented.

Thank you for suggestion. We think that changes between 3.47, 3.06, 3.56 and 3.54 in groups 4_NC, 44_PC, 44_67 and 44_87Y respectively (Table 5) can be considered as bacteriostatic. From our data we cannot claim if found of E. coli belong to commensal or pathogenic strain. Also, we did found Klebsiella pneumoniae in the cecum of birds fed with no additional materials and Enterobacter cloacae was found in group NC as well in the PC. Thus we came with general statement that fermented rapeseed meal acts better than unfermented.

Line 408 IF there are growth data , why don‘t you include them here this would give a more fully picture of the experiment, but to write there was no effect and data not shown is not acceptable..include growth data!!

We have added growth data in Table S1.

Line413… fermented feed additives?? Fermented feed…feed additives have to be registered and usually are given in very low amounts to the diets. By the way as you may notice all these studies used higher amounts of the respective fermented feed. I still haven’t found any explanation or rationale why only 3 % FRSM was added

Thank you for paying attention. In fact, our fermented rapeseed meal is not a feed additive, but a feed material. The text has been corrected accordingly.

The composition of the feed is given in table 1. Nutritional practice and standards say that rapeseed meal should not constitute more than 5% of the feed mix, so we decided that 3% would be a safe value. We added appropriate references in line 83.

Line 446 even the unfermented rapeseed stimulated L. salivarus and had numerical reduction of other bacteria so why do  you recommend only the fermented rapeseed meal...maybe good if you discuss as well the the results of the unfermented RSM

Rapeseed meal is a feed material and could be recommended. However, it is worth remembering about many factors that affect nutritional effects and health. The rapeseed meal contains several anti-nutritional substances, hence recommended limitations. Fermentation mostly solves this problem, but it was not the subject of this publication. L. salivarus was indeed stimulated by URSM. However, fermented rapeseed meal stimulated L. crispatus/johnsonii (44_87Y) and L. reuterii (44_67). We did recommend fermented rapeseed meal just because it acted slightly better. We changed our conclusions in lines 426-432.

Line 447  … there were no Eukaryote measured in this study…your conclusion should be based on the results of present study and not previous or other peoples studies.

Thank you. That is true, Eukaryotes were not measured in this study. However metagenomic data indicates that Blastocystis hominis was present in the cecal material of birds fed with no addition (44_NC) what is shown on Fig. 3. Our conclusions are based only on result that we obtained.